# InsNeXt: Training Scalable Insertion-Based Language Models From Scratch

## Abstract

Insertion-based language models like Insertion Transformer and InsNet have shown promises as strong alternatives to autoregressive models with better inference-time efficiency and controllablility. However, their training-time scalability has been limited by computational inefficiency and obsolete model designs. We aim to tackle this problem with **InsNeXt**, an insertion-based language model architecture integrating recent advancements of language model systems to achieve improved scalability. We scale InsNeXt from 154M up to as large as 0.6B parameters with context window of 4096 by combining sentence-level training and document-level training to better encode the context and bring out the benefits of insertion-based models to encode bi-directional contexts. In addition, we propose a novel context encoding mechanism specialized for insertion-based decoding. The inference-time mechanism sparsely introduces bidirectional re-encoding of context, thus effectively leverages the models' bidirectional context reception while preserving the same level of computational efficiency as conventional autoregressive decoding. We evaluate the pretrained InsNeXt models from the perspective of representation learning, commonsense reasoning and controllable generation. InsNeXt models achieve similar or better performance in comparison to the state-of-the-art similar-sized autoregressive models, making them a class of solid representation learners and powerful controllable insertion-based generators.

## 1 Introduction

Large-scale pretrained autoregressive language models have dominated the paradigm of natural language generation over the past few years. These models, including the GPTs (Radford et al., 2018, 2019; Brown et al., 2020; OpenAI, 2022; Achiam et al., 2023), LLaMAs (Touvron et al., 2023a,b; Dubey et al., 2024; Meta, 2025), Phis (Li et al., 2023a,b; Javaheripi et al., 2023; Abdin et al., 2024), Qwen LLMs (Bai et al., 2023; Yang et al., 2024a,b) and Deepseek LLMs (Bi et al., 2024; DeepSeek-AI et al., 2024; Liu et al., 2024; Guo et al., 2025), have demonstrated impressive training-time scalability and versatile performance across a wide range of tasks. However, their inference-time efficiency and lack of controllablility motivate people to explore alternative methods for pretrained language models.

One potential alternative is the *insertion-based* language model (Stern et al., 2019; Gu et al., 2019; Lu et al., 2022). Unlike autoregressive models, which generate text strictly from left to right, insertion-based models allow tokens to be generated at arbitrary positions in arbitrary order, making them inherently more flexible and better aligned with the compositional nature of human language. Moreover, their potential for improved controllability and fine-grained editing offers unique advantages in tasks requiring structured or context-sensitive generation (Zhang et al., 2020). Despite these benefits, attempts to scale up insertion-based models remain limited, mainly due to their training-time inefficiency especially compared to modern *large language models* (LLMs).

Submitted to 39th Conference on Neural Information Processing Systems (NeurIPS 2025). Do not distribute.

The advent of efficient training techniques and architectural improvements in modern LLMs offers a pathway to overcoming these limitations. Practices such as FlashAttention/Memory Efficient Attention (Dao et al., 2022; Dao, 2023; Shah et al., 2024; Rabe and Staats, 2022; Dong et al., 2024) significantly reduce the memory overhead of attention mechanisms while accelerating computation. Advances in layer normalization (Xiong et al., 2020), optimization techniques (Loshchilov and Hutter, 2017; Rasley et al., 2020a), and data scaling strategies (Hoffmann et al., 2022) have revolutionized the training of large-scale models in other dimensions. These innovations have allowed models with billions of parameters to be trained efficiently (Achiam et al., 2023; Dubey et al., 2024), unlocking new levels of performance in NLP tasks. However, these advancements are mostly specialized for autoregressive LLMs, with little exploration of their applicability to insertion-based models.

In this work, we address these challenges by integrating state-of-the-art practices to forge the insertion-based language model InsNeXt, a modern architecture capable of training-time scaling on par with traditional autoregressive models. By incorporating techniques such as FlexAttention (Dong et al., 2024), improved positional encodings, and optimized training pipelines, we achieve substantial improvements in computational efficiency and scalability.

We pretrain InsNeXt with two major configurations and a few ablative ones, ranging from 154M to 587M parameters, supporting a maximum context window of 4096 tokens. The training is performed under a two-stage fashion: sentence-level pretraining on a BERT (Devlin et al., 2019)-style Wikipedia+books dataset and document-level pretraining on the SlimPajama (Soboleva et al., 2023) dataset. As one of the foundation works, we study on a huge variety of alternative designs in different aspects insertion-based models, revealing arguably the best practices of which. We also propose an improved re-contextualization mechanism for the insertion-based decoding to better utilize the models' bidirectional context reception. The resulting models are evaluated on a broad range of tasks, demonstrating their effectiveness both as bidirectional BERT-style representation learners for natural language understanding (NLU) and as insertion-based decoders for generative and likelihood prediction tasks. We believe these qualities distinguish InsNeXt models from autoregressive counterparts and highlighting their potential to redefine the landscape of language modeling.

## 2 Methodology

### 2.1 Revisiting InsNet with a Contrastive Study against the (Large-Scale) Autoregressive Models

InsNet (Lu et al., 2022) is one of the first works that focus on tackling the training-efficiency issue of insertion-based language models. It addresses the efficiency issues in a practical training of insertion-based models in two folds: during the *context encoding* phase and the *action prediction* phase.

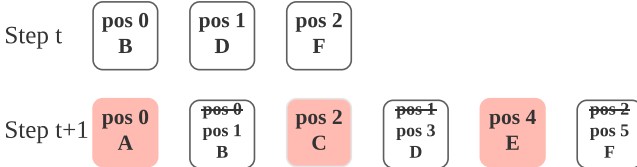

Figure 1: Illustration of the volatile position problem by Lu et al. (2022).

**Context Encoding: the Volatile Position Problem and the Solution** The high parallelizability and efficiency of context encoding in autoregressive transformer-based decoder-only LLMs (Vaswani et al., 2017; Radford et al., 2019) are widely considered some of the most important factors towards their success. Compared to these models, the vanilla insertion-based model Insertion Transformer (InsT) (Stern et al., 2019) falls short. Lu et al. (2022) argues that the biggest training-time performance issue for InsT comes from the position encoding inefficiency, namely the *volatile position* problem. We include their original illustration of the problem as in Figure 1. InsT relies on absolute position embeddings[1] that bind each token's representation to its index in the *partial* sequence. When a new token is inserted, the indices of the existing tokens shift, so their previously cached position embeddings become invalid. To restore consistent key/query vectors for attention, the compute-heavy

---

[1] Alternatively, relative position encodings with static distance, like in XLNet (Yang et al., 2019) and Roformer (Su et al., 2024)

Transformer models need to be run over the entire updated context at every step, incurring a full re-encoding pass per insertion and pushing training complexity to at most $O(n)$ per sequence. InsNet improves insertion-based text generation by introducing an insertion-oriented, relative positional encoding called the *offset* matrix. Offset matrices only concern the *relative* spatial interactions between tokens that are *already* inserted at each step. When a new token is inserted, all previously computed position encodings remain unchanged, and the incoming token reports its pairwise distance to the existing tokens into the new *offset* matrix; this lets the model reuse a single context encoding for an entire sequence, cutting training-time re-encoding from $O(n)$ to $O(1)$ and preserving full distance information. In addition, the authors develop a fast *offset-compression algorithm* that builds the offset matrix from permutation indices with simple masking and in-row ranking operations, avoiding the naive $O(n^2)$ construction cost in the sequential execution. Each transformer layer then uses sinusoidal embeddings to concern the offset values in the attention score, yielding insertion-oriented distance-aware representations without extra passes.

**Action Prediction: Next-Token Prediction (NTP) vs. Next-Insertion Prediction (NiTP)**  In autoregressive decoding, the prediction task is simple: predict the single *next token* (NTP) that will be appended to the current prefix. However, in insertion-based generation, the model must decide at least *both* **where** to insert and **what** to insert. The position choice is scalable and intuitive, because it can be cast as a single-head attention over the sequence, thus benefiting from efficient attention kernels with block-wise reduction (Rabe and Staats, 2022; Dao et al., 2022; Dao, 2023; Shah et al., 2024) for scalability. However, predicting the *next-inserted token* (NiTP) is harder. In an autoregressive decoder-only Transformer, given a training sequence, the mapping from a partial prefix to its next token is deterministic, so no extra aggregation is needed beyond the final layer. By contrast, in insertion-based generation, the representation for NiTP changes with every candidate slot, even under the same prefix. InsNet follows InsT and formalizes each candidate position as an *insertion slot*. It then builds the *slot representation* embeddings with a lightweight sparse attention over last-layer hidden vectors: the **left** and **right** neighbors, and the **most recently** inserted token.

## 2.2 InsNeXt: Scalable Insertion-based Language Model with Advanced Transformer Designs

InsNet inherits a lot of elements from XLNet (Yang et al., 2019), a BERT-era model with legacy designs, many of which are later proven to be suboptimal in larger scale pretraining (Brown et al., 2020; Touvron et al., 2023a). We hereby discuss both the recent advances for decoder-only transformers that we adopt to scale up the proposed model InsNeXt, and new model designs that we craft with originality to facilitate the scalability of InsNeXt.

**General Design: the Major Model and the Ablated Variants**  Since the scaling law and best practice of such scalable insertion-based language models remains hugely underexplored, we conduct a fairly broad range of model design ablation in each of the aforementioned aspects. However, it is too computationally expensive to exhaust all of the possible combinations of designs. Thus, we first select a basic, safe combination of model designs to build a basic model for the ablation study, then combine all best practices we found in the model design study to build the final major models we deliver. The basic model is a 12-layer, 12-headed transformer with InsNet-style sinusoidal relative position encoding. It has a dimensionality of 768, and GeLU pre-LN FFN layers with intermediate size of 3072. It uses *untied* input and output embeddings. The total parameter count is 171M. It uses shallow aggregation for NiTP.

Provided our computation limitations, we present the two major setups in scaling up insertion-based models: the *base*-sized 154M model and the *advanced*-sized 573M model. InsNeXt-base is a 16-layer, 12-headed transformer model with insertion-oriented ALiBi as the position encoding. It has a hidden size of 768 and SwiGLU FFN layers with intermediate size of 2048. It uses *tied* input and output embeddings. InsNeXt-advanced shares most architectural designs with the base-sized model with only size expansion. It has 32 layers of 18-headed attention, and a hidden size of 1152. It uses SwiGLU FFN layers with intermediate size of 3072. Both models use deep aggregation for NiTP.

**Residual Connection and Layer Normalization**  In contrast to InsNet's legacy PostLN block, we conduct an ablation study over a few more recently proposed normalization alternatives, including pre-normalization (pre-LN) (Xiong et al., 2020), two-hop pre-LN proposed in MEGALODON (Ma et al., 2024) and Deepnorm (Wang et al., 2024). Results show that most recently published normalization blocks yield observably better stability and scalability than the legacy post-LN block, which is consistent with their reported performance in autoregressive models and encoder models. For an

ablative and contrastive study on the effect of warmup iteration numbers under each normalization choice, please refer to the appendix A.4.

**Position Encoding**  Scalable Insertion-based language models are inherently incompatible with absolute position encoding and relative position encoding with static distance assumptions. This is a direct result of the volatile position problem in insertion-based generation. Thus, common relative position encodings, *e.g.* the Rotary Position Encoding (RoPE) (Su et al., 2024) and T5-bias (Raffel et al., 2020), are not directly applicable without modification. We explore other alternatives, including the original InsNet sinusoidal relative position encoding and ALiBi (Press et al., 2021), as they both directly model the *interaction* of different positions and thus can be altered to reflect the insertion-oriented position layout. For more information on the attempts at modifying and efficiently implementing the two position encodings, please refer to the appendix A.2.

**Slot Aggregation: Deep Aggregation using Two-stream Attention**  In our early attempt to scale up the model, we find that while the shallow aggregation proposed in InsNet is efficient and capable enough for smaller models on small datasets, it is no longer the best practice when both the training data and model sizes increase. A notable observation is that during large-scale pretraining, the model should be aware of the location of insertion after permutation. To better utilize the potential of increasing data and model capability during scaling-up, we follow the practice in XLNet and adopt the two-stream attention mechanism to *deep aggregate* the slot representation from layers of the context encoding. A detailed illustration of the insertion-oriented two-stream attention in both training and decoding can be found in the appendix Figure 3.

**Position Prediction: Soft-capped Position Logits**  In actual language usage, when humans recursively refine or expand a sentence, it is possible that there are multiple correct slots for the next step of expansion. The original position-predicting attention in InsNet does not reflect this. In InsNeXt, we introduce a soft-cap mechanism to the position logits to encourage the model to learn a uniform distribution over all feasible next-insertion slots. Given the soft-caps $K > 0$, the modified position logit $-K < a_{pos}^K < K$ is computed by $a_{pos}^K = K \cdot \tanh(\frac{a_{pos}}{K})$.

## 2.3  Training Details

### 2.3.1  Dataset Preparation

Due to the intractable nature of exhaustive enumeration of all permutations for longer sequences, we conduct the training of InsNeXt models in a two-stage fashion: sentence-level training on a BERT-style dataset and document-level training on the first 60B of the SlimPajama dataset. A study in the earlier stage of our attempt shows that this is beneficial for the model to converge faster, compared to directly training on the document-level data.

**Sentence-level Data**  The sentence-level data is crafted from a mixture of Wikipedia-English-2023, the Gutenberg Project dataset (PGLAF, 1971) and the BookCorpus dataset (Zhu et al., 2015). Only natural sentences that start with alphabetical characters and end with terminating punctuation are selected. Two consecutive sentences that appear in the same document are concatenated into a single training sequence for the model to learn the basic concept of *moving on* to the next sentence when one sentence is finished. The maximum sequence length is set to be 128, and models are trained to predict the likelihood of at most only the first 96 insertion operations to avoid overfitting to only complete sentence pairs.

**Document-level Data**  The document-level data is crafted from the SlimPajama dataset (Soboleva et al., 2023). The original SlimPajama dataset consists of 627B tokens, of which we take the first 60B tokens to facilitate our training process. Due to the limit of computational resources, in the first 95% batches of training, each sequence has a token limit of 1024, and we use only the last 5% for context length expansion to at most 4096.

### 2.3.2  Tokenization and Permutation of Insertion Operations

**Tokenizer**  Following the setup of prior autoregressive LMs like Pythia (Biderman et al., 2023), OlMo (Groeneveld et al., 2024; OLMo et al., 2024) and ModernBERT (Warner et al., 2024), we use the GPTNeoXTokenizer with a vocabulary size of 50254. The only notable modification is that we force the tokenizer to split each digit in numbers.

**Permutation of Insertion Operations**  To ensure the integrity of each natural word, the tokens within the same natural word are always grouped together and generated/predicted autoregressively in the permutation. We argue that it is mostly only reasonable to assume the recursive structure and compositionality in the same natural sentence. Thus, for inter-word permutations, all permutations are limited to within natural sentences chunked by the SpaCy (Honnibal et al., 2017) sentencizer. We acknowledge that this is an potentially problematic implementation, especially for those non-text, Markdown/HTML script data in SlimPajama dataset. We leave the study of a more principled, domain-agnostic permutation algorithm for future work.

**Interleaved PrefixLM Masking**  We adopt a partially prefix-LM (Raffel et al., 2020) paradigm of training to encourage the model to learn a representation that captures bidirectional information and can be obtained even with attention mask removed. In sentence-level training, we have a 50% chance to remove part of the attention mask for a uniformly random proportion of the first few insertion operations. In document-level training, the proportion of training sequences with this prefix-LM masking is reduced to 10% to avoid sparsity in token prediction and computation utilization. In practice, this helps improving InsNeXt robustness as a representation learner, and even brings us the possibility to resolve the distribution shift issue during decoding of insertion-based models. We will discuss the this special masking with more details and illustrations in the appendix A.3.

### 2.3.3 Optimization Setup

**Batch Size and Distributed Training**  In both stages of training, we use a global batch size of 1M tokens for all configurations of the model unless otherwise stated. For more information, please refer to our appendix in A.3.

**Optimizer and Learning Rate Schedule**  We use the AdamW (Loshchilov and Hutter, 2017) optimizer with gradient norm clipping of 1.0 and beta values of (0.9, 0.9). During sentence-level training, following the practice of ModernBERT (Warner et al., 2024), we use a trapezoidal learning rate scheduler with 10000 warmup steps and a peak learning rate of 4e-4. After 110000 further iteration steps of constant LR training, we cooldown the model in 60000 iteration steps by cosine-decaying the learning rate to 1e-4. For document-level training, we linearly warmup the model within the first 1000 steps and then fine-pretrain the model with a constant learning rate of 1e-4.

**Initialization and Other Important Details**  For all experiments, we choose the soft position logit cap $K = 3$. Following the practice of many recently published LLMs, we initialize all parameters from a normal distribution with stddev $= \sqrt{\frac{2}{5D}}$ ($D$ is the hidden size), unless otherwise stated.

### 2.4 Decoding

We hereby briefly explain the details on how to decode from a pretrained InsNeXt model.

**Position Selection**  Unlike autoregressive models that only need to perform next-token prediction, the proposed scalable insertion-based model needs to first determine the decoding position and then aggregate the slot representations in each step of the decoding. The selection of the position can be performed deterministically by taking the argmax among all position logits, or stochastically by sampling from the softmax-ed distribution $p_{pos} \propto \exp(a_{pos}^K)$ over all positions. Note that one can always choose multiple ones of the top-N ($N > 1$) positions distribution simultaneously and attempt to decode in parallel as in InsNet (Lu et al., 2022). We leave the application of this feature for future work and only focus on sequential (one token at a time) decoding in this paper.

**Slot Aggregation for NiTP**  After the next position prediction, we generate the offset matrix and apply the two-stream attention to compute the representation for NiTP. Note that in actual deployment environment, this step can be significantly accelerated by caching KVs of the previous context encoding steps as in decoder transformers. We then take the deep-aggregated slot representation and project it using the transposed word embedding matrix to generate a distribution over the vocabulary. Any decoding algorithms or logit modifiers that work for autoregressive models can then be applied here without much adaptation.

**Efficient Bidirectional Re-contextualization**  The original way of using the insertion-oriented position encoding and upper-triangular masking to efficiently model the spatial relation in insertion-based generation has caused two major issues. The first one is a performance issue similar to the *exposure bias* in autoregressive models. In a uni-directionally encoded insertion-based generation process, the spatial relation depicted by the interaction of the first few tokens can be inaccurate

especially if we consider longer bidirectional context. This causes the model's internal error to accumulate over time, and eventually lead to degenerated performance on the long run.

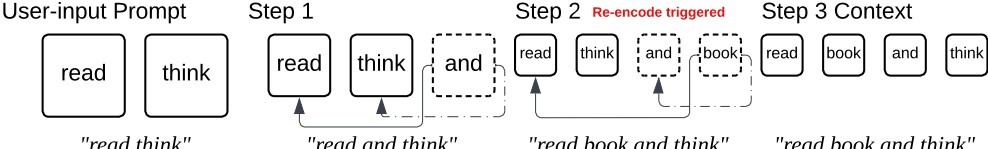

Figure 2: Illustration of when the bidirectional re-encoding happens in the proposed Efficient Bidirectional Re-contextualization. Solid line token blocks are ones encoded with a bidirectional attention; dashed line ones are encoded using the unidirectional efficient offset matrix that does not concern future insertions.

The second issue is about a proper permutation assumption for a user-input prompt. The permutation of insertion operations has an underlying effect on how the model understands the structuring of the generated contents. For model-generated contents, we always first predict the position and then the token, so the permutation is naturally present. However, this is not the case for user-input prompts. In previous insertion-based models without bidirectional encoders like InDIGO (Gu et al., 2019) and InsNet (Lu et al., 2022), a common practice is to simply assume an autoregressive permutation or randomly generate the permutation. However, this either causes train/test discrepancies or injects additional stochasticity to the context encoding, which eventually harms the reliability and controllability of the model. One essential solution is to switch to fully bidirectional context encoding as in InsT (Stern et al., 2019) and some recently published diffusion language models . However, this is inefficient, as fully re-encoding upon any context update triggered by new insertions is extremely compute-expensive.

Inspired by C++ STL's implementation of the vector container (see analysis in Cormen et al. (2022)), which allows dynamic reallocation of memory for a growing array that supports random access and appending, we propose an efficient bidirectional re-contextualization mechanism that solves the two issues of the uni-directional context encoding at once. The idea is surprisingly simple - for user-assigned input or the initial empty string context, we simply encode the sequence bidirectionally by removing the lower-triangular attention mask. We continue to generate the next few tokens with unidirectionally encoded context updates while retaining the bidirectionally encoded part in the prefixLM paradigm (Raffel et al., 2020), until the generated tokens since last re-contextualization surpass the length of the bidirectionally encoded section. Then, we discard all KV cache so far (if any) and an expensive yet spatially unbiased fully bidirectional context re-encoding will be triggered. It is expected that: 1) at least half of the context is bidirectionally encoded, so that the spatial relation won't be significantly and irreversibly corrupted by insertion operations, 2) the expensive fully-bidirectional re-encoding is not triggered very frequently. In fact, one can easily prove that the expected computational overhead of a decoding process with the proposed re-contextualization is at the same scale of the vanilla uni-directional one with only a marginal extra cost.

## 3 Experiments

We conduct our experiments in three major parts:

- **Natural Language Understanding Tasks** experiments aim to test the basic generalizability and quality of the learned representation. We also reuse this experimental setup in our ablative model design exploration to find the model design best practices.

- **Commonsense Reasoning Tasks** experiments aim to test whether the pretrained model is able to capture the world model knowledge in its parameters, and induce a stronger preference on the commonsensical continuation over the other ones;

- **Controllable Generation Tasks** experiments are conducted to show the unique controllability hegemony of insertion-based language models over traditional autoregressive models.

## 3.1 Experiment 1 - Representation Learning Study using Discriminative Natural Language Understanding Tasks

Following previous works on representation learning, we focus on two subtasks SST-2 (Socher et al., 2013) and MNLI (Williams et al., 2018) from the GLUE (Wang et al., 2018) leaderboard. We argue that these two tasks reflect the pretrained model's learned representation's generalizability for single sentence (represented by SST-2) and multi-sentence (represented by MNLI) scenarios well.

We first explore the best scalable designs for insertion-based language models. We start from a *basic* model that reproduces most architectural designs as InsNet, then replace different components of the model and train the altered variants. Due to the limit of computational resources, we only train the basic model and the altered variants on the sentence-level data for 120000 iterations without the trapezoidal cooldown. For more details, please refer to the appendix.

We report the evaluation on the development split of SST-2 and matched version of MNLI. We compose the best practices of each module and train the major models evaluated in all following experiments. A more comprehensive study of the major models' performance on the GLUE leaderboard, and a detailed ablation of different model designs can be found in the appendix.

Table 1: Representation learning study in comparison with the baseline models. The best performance of each category/group is marked with **bold** font and the notable second place winner is marked with underline. We consider both the accuracy on the downstream tasks and training efficiency for architecture selection. All results are reported as the average of models with 3 different random seeds.

| Model Variant | #Params | SST-2 | MNLI-m |
|---|---|---|---|
| **Baselines - Generative** | | | |
| GPT-2-base | 124M | 91.85% | 81.23% |
| GPT-2-medium | 355M | 92.09% | 85.23% |
| Pythia-160m | 123M | 89.30% | 78.96% |
| Pythia-410m | 354M | 91.55% | 83.03% |
| Pythia-1b | 908M | 91.66% | 83.85% |
| T5-Small | 61M | 90.44% | 82.07% |
| T5-Base | 223M | 92.54% | **85.30%** |
| Qwen2.5-0.5b | 494M | **94.26%** | 84.65% |
| **Baselines - Discriminative** | | | |
| BERT-base | 108M | 92.27% | 84.14% |
| BERT-large | 334M | 93.73% | 85.66% |
| RoBERTa-base | 125M | 94.26% | 87.43% |
| RoBERTa-large | 355M | **95.94%** | **90.26%** |
| InsNet (reproduced) | 171M | 91.85% | 82.20% |
| InsNeXt-base (Ours) | 154M | 93.00% | 83.23% |
| InsNeXt-advanced (Ours) | 573M | 94.15% | 85.94% |

**Discussion**  Results show that the proposed InsNeXt models are solid representation learners, especially compared to peer generative models, while all of which still fall short against SOTA encoding-oriented models like RoBERTa, even with doubled size like TinyLLaMa (Zhang et al., 2024).

## 3.2 Experiment 2 - Zero-shot Commonsense Reasoning Tasks

Following the practice of TinyLLaMa (Zhang et al., 2024), we conduct a commonsense reasoning study of our model against a fair range of popular small-to-medium-sized language models in the community. We choose the HellaSwag (Zellers et al., 2019), Obqa (Mihaylov et al., 2018) and WinoGrande (Sakaguchi et al., 2019) datasets as our testbed. The models' choice are selected using length-normalized likelihood scores, following prior practices. Note that the commonsense reasoning tasks here are likelihood-predictive ones rather than generative ones. We will discuss the generative commonsense reasoning task CommonGen in the next section.

Table 2: Commonsense Reasoning Evaluation.

| Model | #Params | HellaSwag | Obqa | WinoGrande |
|---|---|---|---|---|
| Random Selection | - | 25.00% | 25.00% | 50.00% |
| GPT-2-base | 124M | 31.14% | 27.20% | 51.62% |
| GPT-2-medium | 355M | 39.38% | 30.20% | 53.20% |
| Pythia-160m | 123M | 30.17% | 27.00% | 51.30% |
| Pythia-410m | 354M | 40.52% | 29.40% | 53.04% |
| Pythia-1b | 908M | 47.10% | 31.40% | 53.43% |
| Qwen2.5-0.5b | 494M | **52.17%** | **35.20%** | **56.59%** |
| InsNet (reproduced) | 171M | 27.39% | 24.40% | 50.71% |
| InsNeXt-base (Ours) | 154M | 33.47% | 30.20% | 52.37% |
| InsNeXt-advanced (Ours) | 573M | 53.63% | 34.80% | 55.87% |

**Discussion** Results show that the proposed InsNeXt models are trained well enough to compress the world knowledge in the training data into its parameters, as a result having similar commonsense reasoning capabilities to similar-sized autoregressive models.

## 3.3 Experiment 3 - Controllable Generation Tasks

We experiment with the lexically-controlled generation tasks following the setup of InsNet to measure the performance of the proposed InsNeXt model. We examine the evaluated models on the Yelp 160K, WMT News and CommonGen datasets. Note that insertion-based models are not directly able to handle fuzzy lexical constraints that allow reflections. InsNet reports performance using the original form of the concept words without considering reflections. In this work, we train the models as insertion-based decoders that take in full keyword sequences as input, first fulfill the lexical constraints with correct reflections and then generate the rest of the context. This significantly improves their coverage of keywords, at the cost of slightly imperfect keyword coverage rates compared to some recent search-based and HMM-based methods like GeLaTo (Zhang et al., 2023).

Table 3: Controllable Generation Evaluation. Both BLEU-4 and lexical constraint coverage (shown with BLEU-4↑/Coverage↑) are reported. Note that the Yelp and News are high-variance datasets, so it's natural that all models have rather lower BLEU-4 score compared to that in CommonGen.

| Model | #Params | Yelp 160K | News | CommonGen-dev |
|---|---|---|---|---|
| GPT-2-base | 124M | 5.83/77.50% | 6.54/63.30% | 22.90/65.4% |
| GPT-2-medium | 355M | 6.99/87.70% | 7.56/80.60% | 24.65/83.2% |
| Pythia-160m | 123M | 5.83/52.80% | 5.50/47.00% | 20.97/67.4% |
| Pythia-410m | 354M | 7.29/85.10% | 6.64/72.00% | 22.29/86.2% |
| Qwen2.5-0.5b | 494M | 7.46/90.30% | 7.42/92.40% | 23.37/92.0% |
| Insertion Transformer (BERT init+POINTER) | 357M | 3.79/100% | 3.04/100% | 16.70/97.9% |
| InsNet (Original Report) | 171M | 5.78/100% | 4.96/100% | 18.71/100% |
| InsNet (reproduced, w/ keyword-first perm.) | 171M | 5.63/100% | 4.87/100% | 21.36/98.3% |
| InsNeXt-base | 154M | 6.73/100% | 6.60/100% | 23.46/97.9% |
| InsNeXt-advanced | 573M | **8.13/100%** | **8.01/100%** | **25.73/98.7%** |

**Discussion** Results show that the proposed InsNeXt models are more controllable and capable language generators than its autoregressive counterparts and prior insertion-based models, in term of both constraint coverage (near-perfect to perfect) and overall generation quality.

## 4 Related Works

**Insertion-based Language Models** Insertion-based language models offer a flexible alternative by constructing sequences through token insertions, in contrast to the conventional left-to-right continuation in traditional autoregressive LMs. The Insertion Transformer (Stern et al., 2019) is an iterative, partially autoregressive model that generates sequences based on insertion operations.

Building upon this, POINTER (Zhang et al., 2020) was developed for hard-constrained text generation, progressively inserting tokens in a parallel manner to complete sequences efficiently. To enhance training efficiency and decoding flexibility, InsNet (Lu et al., 2022) introduced an insertion-oriented position encoding and a lightweight slot representation strategy, enabling both parallel and sequential decoding. Further advancements include the design of Fractional Positional Encoding (FPE) for Insertion Transformers (Zhang et al., 2021), allowing the reuse of representations from previous steps, yet introducing extra discrepancy between training and decoding.

**Relative Position Encodings**   Recent advancements in transformer models have led to various methods for encoding positional information. Transformer-XL (Dai et al., 2019) and XLNet (Yang et al., 2019) introduced relative positional embeddings to better capture dependencies across long sequences. T5 (Raffel et al., 2020) employs learnable relative position biases, enhancing its adaptability to different sequence lengths. ALiBi (Attention with Linear Biases) (Press et al., 2021) adds linear biases directly to attention scores, facilitating the processing of longer sequences. Rotary Position Embedding (RoPE) (Su et al., 2024) integrates positional information through rotation matrices applied to token embeddings. Most of these techniques are tailored for fixed-order sequence modeling and may not be directly applicable to insertion-based generation. Notably, ALiBi and T5's relative position biases are exceptions; their designs involve rectifiers as a function of *only* the offset between positions, making them potential for insertion-oriented scenarios. However, T5 bias faces scalability issues as no mainstream efficient kernels support training with it yet.

**Efficient Scaled Dot Product Attention**   In recent years, the development of memory and compute-efficient attention mechanisms has seen notable advancements. Initially, Rabe and Staats (2022) proposed a method to reduce memory requirements from quadratic to linear scale by partitioning attention computations into smaller blocks, fitting within a GPU's on-chip memory. Building upon this, the FlashAttention algorithm (Dao et al., 2022) further optimized efficiency by managing data movement and computation to minimize memory usage and increase computational speed. FlashAttention-2/3 (Dao, 2023; Shah et al., 2024) continue to incorporate hardware-related optimizations. More recently, FlexAttention (Dong et al., 2024) has been developed to combine the flexibility of reprogrammable attention score modifiers with the performance benefits of FlashAttention, allowing researchers to experiment with various attention variants efficiently.

# 5   Conclusion and Future Work

We present InsNeXt, a modern-level scalable insertion-based language model integrating recent advances in LLMs. We explore and discuss model design alternatives of each component for insertion-based generation, the basic scaling pattern of the model, as well as the best training strategies from scratch for them. We then follow the best practices from the architecture study to train two major specifications of the model, one (InsNeXt-*base*) with 154M parameters and another (InsNeXt-*advanced*) with 573M parameters. We evaluate the fully renovated insertion-based models under three major scenarios: 1) representation learning capabilities using NLU tasks; 2) commonsense reasoning capabilities using HellaSwag, WinoGrande and Obqa; and 3) controllable generation on Yelp, news and CommonGen datasets. Results show that the InsNeXt models are of competitive performance with state-of-the-art autoregressive language models, while preserving their unique advantages especially in terms of controllability.

We are aware of the recent progress in discrete-space diffusion language models with autoregressive LLM warmups. These models have the potential to provide parallelizability of generation as well as heterogeneous allocation and assignment of computational power for tokens with different level of predictability. Existing Diffusion LLMs are mostly based on the MLM objective, meaning it has to pre-determine the number of placeholder tokens and then learn to fulfill them. We argue that this is a flawed solution, and InsNeXt can be viewed as the prototype model towards a more flexible, powerful family of diffusion language models as the foundation for the next generation of artificial intelligence.

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

# A  Supplementary Details of Model and Algorithm Designs

## A.1  Additional Model Design Information

**Activation**  In addition to the legacy GeLU (Hendrycks and Gimpel, 2016) feed-forward layers, we also consider SwiGLU, which is a variant of the Gated-Linear Units (Shazeer, 2020) that reportedly provides better parameter efficiency in more recently published LLMs. Note that, to facilitate training efficiency and provide a fair comparison, we intentionally choose model dimensionality divisible by 3. This way, all matrix/vector multiplication operations can be performed on tensors with dimensionality of multiples of 128/64 for best TensorCore practices, without altering the total number of parameters compared to its GeLU counterparts.

## A.2  Implementing Position Encoding with Efficient Attention Kernels

We hereby discuss the related attempts we make to implement the mentioned position encodings efficiently.

**InsNet-style Sinusoidal Relative Position Encoding**  For the original InsNet sinusoidal relative position, no off-the-shelf efficient attention kernel is available at the moment when this paper is submitted, so we build our own memory-efficient PyTorch attention kernel with eager-mode code, and then transform it to the Triton implementation of the attention mechanism to include the block-wise reduction tricks that other efficient attention kernels provide. See code snippet 1.

**Insertion-oriented ALiBi**  For insertion-oriented position encoding with ALiBi, we make a minor modification of it on the denominator of the bias rescaling factor $\alpha_i = \frac{1}{2^{k_i}}$. Instead of using uniformly interpolated exponentials from 1.0 to 8.0 in each attention head, as suggested by the original ALiBi paper, we simply choose $k_i$ to be the 1-based attention head index. For example, an attention layer with 12 attention heads will have $\alpha$ to be $\left[\frac{1}{2^1}, \frac{1}{2^2}, \frac{1}{2^3}, ..., \frac{1}{2^{11}}, \frac{1}{2^{12}}\right]$, instead of $\left[\frac{1}{2^{(2/3)}}, \frac{1}{2^{(4/3)}}, \frac{1}{2^2}, ..., \frac{1}{2^{(22/3)}}, \frac{1}{2^8}\right]$ as in vanilla ALiBi. For larger models with more dimensions, this simply adds in extra attention heads that have slower bias-term decay when relative distance increases. This helps the larger model to utilize the additional capacity to handle longer-range dependencies or generate less position-sensitive perceptions without significantly changing its behavior in short-range dependencies. In short-range dependencies with attention heads that have larger bias decay denominators, it is approximately equivalent to omitting position information in some of the model dimensions, which is reportedly a common practice in recent position encodings like RoPE. During training, InsNeXt with insertion-oriented ALiBi is implemented with the FlexAttention (Dong et al., 2024) kernel, which is a Triton-implemented FlashAttention-2/3 alternative that facilitates scalable training of flexible attention modifier. However, up to the time of this submission, the inference-friendly version of FlexAttention is still under-optimized. During inference, we simply fall back to Memory Efficient Attention as our attention kernel choice.

## A.3  Pretraining

**Batch Size and Distributed Computation**  We train our model on a single NVIDIA H100x8 SXM node. Unfortunately, we encountered a hardware failure in the early stage of this project that the NVSwitch bridge between the first to the rest of the GPUs was compromised. As a result, we are only able to deploy a 7-way distributed training for the major experiments instead of an 8-way one. In sentence-level training, we use an equivalent batch size of 1536 per GPU along with the gradient accumulation trick, resulting in a global batch size of 10752 sequences and approximately 1M tokens. In the major part of document-level training, we use an equivalent batch size of 144 per GPU, resulting in a global batch size of 1008 sequences and approximately 1M tokens too. We reduce the batch size and learning rate accordingly for the context extension phase to keep the per-batch token number to be around 1M. We build our distributed training script using a composition of the DeepSpeed (Rasley et al., 2020b) and PyTorch (Paszke et al., 2019) libraries, adopting the ZeRO offload stage of 1 for the base-sized model and 2 for the advanced-sized model. We also use a mixed precision training, with the BFloat16 datatype for in-layer gradients, and TensorFloat32 for gradient reduction, accumulation and parameter storage.

**Interleaved PrefixLM Masking**  We adopt a partially prefix-LM (Raffel et al., 2020) paradigm of training to encourage the model to learn a representation that captures bidirectional information and

can be obtained even with attention mask removed. Specifically, we remove part of the attention mask for a uniformly random proportion of the first few insertion operations, so that each token can attend to these unmasked tokens from both directions. See Figure 4.

## A.4 Extensive Ablation Study and Model Design Exploration

In this section we report the experiments we conduct that are intended for ablation study and model design exploration purposes. We start from a reproduced InsNet architecture, replace some of the components, pretrain the resulting model on the aforementioned sentence-level data then test it on the representation learning capabilities as a reflection of model design soundness.

**Position Encoding** We first explore the trade-off between model capability and training scalability of different insertion-oriented position encoding. We additionally report the training throughput at a context length of 1024 tokens per sequence. See the results in 4. All available designs yield similar performance, with T5 Bias slightly outperforming the counterparts. However, as training with T5 bias is not yet supported by any efficient attention kernels, we choose **insertion-oriented ALiBi** as our major position encoding. We leave the implementation of a potentially more capable insertion-based model with T5 bias for future work.

Table 4: Position encoding study results.

| Model Variant | #Params | SST-2 | MNLI-m | #Tokens/s/Chip @1K Cxt. |
|---|---|---|---|---|
| InsNet (reproduced, @Iter. #120K) | | | | |
| - Sinusoidal Position Embedding | 171M | 91.85% | 82.20% | 17384 |
| - T5 Bias | 165M | 92.01% | 82.73% | 32331 |
| - Insertion-oriented ALiBi | 164M | 91.74% | 82.15% | **116144** |

**Residual Connection and Layer Normalization** We hereby explore the stability and model capability differences impacted by the choice of different layer normalization and/or residual connection choices. We consider four major variants: the original post-LN, pre-LN, two-hop MEGALODON pre-LN and DeepNorm. For DeepNorm, we follow their guidance to change the initialization of each layer accordingly. In addition to the final performance report with 10K warmup steps, we also report the performance with less warmup steps, at 0, 100, 1000 respectively. Results are shown in Table 5.

Table 5: Residual connection and layer normalization study results.

| Model Variant | SST-2 | MNLI-m |
|---|---|---|
| InsNet (reproduced, @Iter. #120K) | | |
| - post-LN | | |
| w/ 0 Warmup | Divergence | Divergence |
| w/ 100 Warmup | 89.33% | 79.83% |
| w/ 1000 Warmup | 90.14% | 80.36% |
| w/ 10K Warmup | 91.85% | 82.20% |
| - pre-LN | | |
| w/ 0 Warmup | 90.36% | 81.37% |
| w/ 100 Warmup | 90.59% | 82.25% |
| w/ 1000 Warmup | 91.28% | 81.99% |
| w/ 10K Warmup | 91.74% | 81.93% |
| - two-hop pre-LN | | |
| w/ 0 Warmup | 91.28% | 81.76% |
| w/ 100 Warmup | 91.51% | 81.79% |
| w/ 1000 Warmup | 91.63% | 81.72% |
| w/ 10K Warmup | 91.63% | 81.88% |
| - DeepNorm | | |
| w/ 0 Warmup | 90.71% | 80.62% |
| w/ 100 Warmup | 90.48% | 81.20% |
| w/ 1000 Warmup | 91.51% | 81.47% |
| w/ 10K Warmup | 91.63% | 81.39% |

With 10K warmup steps, all variants yields similar performance, with the legacy post-LN slightly winning by a margin. However, if we adopt fewer warmup steps, the stability of post-LN is significantly compromised, while others are not very sensitive. Notably, we observe faster convergence with two-hop pre-LN with slightly degenerated final performance, and don't observe any advantages of DeepNorm against other alternatives. We acknowledge the possibility that these two layer normalization implementations are more specialized for even larger models, thus still worth trying if we further scale up the training. For our major configurations of the pretraining, we choose **pre-LN** as our layer normalization block choice.

**Slot Aggregation: Deep Aggregation vs. Shallow Aggregation** We concern both the representation learning and generative capabilities impacted by different slot aggregation methods. We train two variants of the *major* model InsNeXt-base under the same setup but with different aggregation method, and test their respective performances on both groups of our experiments. Results are shown as follows:

Table 6: Slot aggregation ablation study results

| Model | SST-2 | MNLI-m | Yelp 160K | News | CommonGen-dev |
|---|---|---|---|---|---|
| InsNet (Original Report) | - | - | 5.78/100% | 4.96/100% | 18.71/100% |
| InsNet (reproduced) | 91.85% | 82.20% | 5.63/100% | 4.87/100% | 21.36/98.3% |
| InsNeXt-base | | | | | |
| w/ shallow agg. | 92.66% | 83.07% | 5.86/100% | 5.13/100% | 22.70/98.1% |
| w/ deep agg. | **93.00%** | **83.23%** | **6.73/100%** | **6.60/100%** | **23.46/97.9%** |

In general, deep aggregation helps the model to better utilize model capability which consequently helps both representation learning and NiTP. We modestly argue that this will be more signficant for even larger models, but we don't have sufficient empirical results to support this claim. The only clear observation from the current results is that the usage of deep aggregation has a slightly more remarkable impact on generation/NiTP than representation learning.

## A.5 Illustration of the Insertion-oriented Two-stream Attention and PrefixLM Masking

Derived from XLNet's illustration of the two-stream attention, we hereby illustrate how insertion-oriented two-stream attention works in InsNeXt. Assume we've generated a permutation $\pi_i$ of all tokens $x_i$, the two streams of attention can be performed as shown in the following figures:

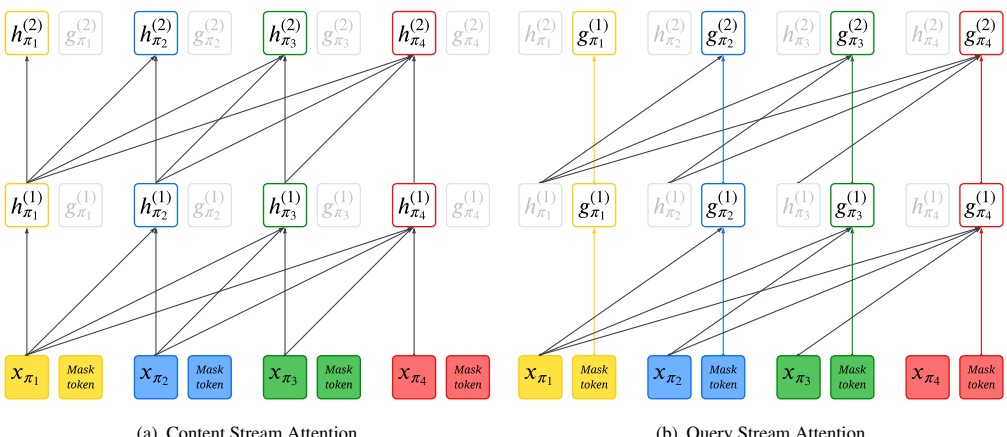

(a) Content Stream Attention

(b) Query Stream Attention

Figure 3: Illustration of the two-stream attention. The content-stream attention aims to model the interaction of context tokens while the query-stream attention aims to aggregate information for NiTP.

Here, $g_{\pi_i}^{(j)}$ means the deep-aggregated representation for the $i$-th inserted token in the sampled permutation $\pi$ from the $j$-th layer of the InsNeXt model.

We also illustrate the prefixLM masking adopted in our pretraining. Given a sequence of $N$ tokens, we uniformly sample $M \sim \text{Uniform}(0, N)$ and remove this part of the lower-triangular attention mask, as illustrated in the following figure:

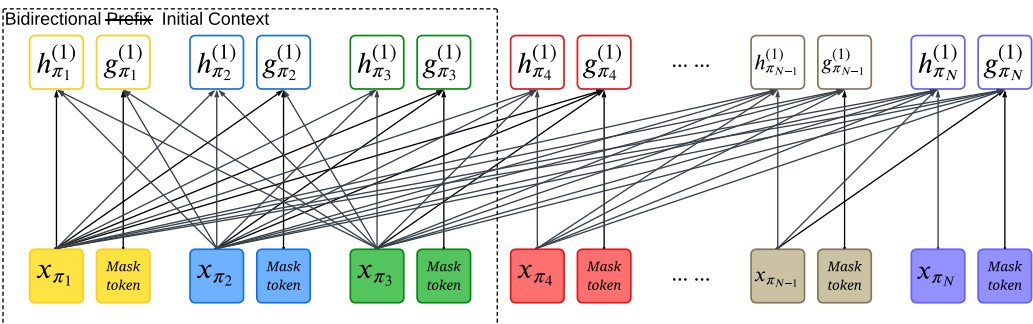

Figure 4: Illustration of the prefixLM style attention masking given a sequence with $N$ tokens and a sampled bidirectional portion $M = 3$. Note that we stop using the notion *prefix* as in insertion-based generation, the bidirectional part is not necessarily the natural prefix, but simply the bidirectionally perceived partial context.

## B  Code Snippet of the Eager-mode Implementation of Sinusoidal Insertion-oriented Position Encoding

We hereby include a few of the important code snippets as more concrete description of the proposed algorithm. To facilitate the readability of code, we report the eager-mode version of the implementation. It can be transformed into an equivalent Triton implementation fairly easily.

Code 1: The eager mode implementation for the memory efficient attention with sinusoidal insertion-oriented position encoding.

```python
def attn_core_logsumexp(query_chunk, key_chunk, value_chunk,
                        mask_chunk=None, offset_chunk=None,
                        query_r=None, key_r=None, offset_clip_range=128):
    attn_scores = torch.einsum('bnqd,bnkd->bnqk', query_chunk, key_chunk)
    attn_scores = attn_scores / math.sqrt(value_chunk.shape[-1])

    if mask_chunk is not None:
        attn_scores = attn_scores + mask_chunk.unsqueeze(1)
    if offset_chunk is not None:
        offset_chunk = offset_chunk.clamp(min=-offset_clip_range, max=offset_clip_range)
        indexed_offset = offset_chunk + offset_clip_range
        attn_scores_qbias = torch.einsum('ind,bnkd->bnik', query_r, key_chunk)
        attn_scores_kbias = torch.einsum('bnqd,jnd->bnqj', query_chunk, key_r)

        # Use the offset to index into the attn_scores_qbias
        # Expand indexed_offset dimensions to match attn_scores_qbias
        indexed_offset = indexed_offset.unsqueeze(1).expand(-1, query_r.size(1), -1, -1)

        # Gather the values using the computed offset index
        attn_scores_qbias = torch.gather(attn_scores_qbias, dim=2, index=indexed_offset) / math.sqrt(
            value_chunk.shape[-1])
        attn_scores_kbias = torch.gather(attn_scores_kbias, dim=3, index=indexed_offset) / math.sqrt(
            value_chunk.shape[-1])
        attn_scores = attn_scores + attn_scores_qbias + attn_scores_kbias

    # Compute logsumexp for numerical stability
    attn_scores = attn_scores.to(torch.float32)
    attn_weight = attn_scores.logsumexp(dim=-1) # bnq
    attn_distro = attn_scores.softmax(dim=-1)

    # Compute softmaxed attention weights without storing attn_scores
    # Compute weighted sum of values
    chunk_reduced_value = torch.einsum('bnkd,bnqk->bnqd', value_chunk, attn_distro.to(value_chunk.dtype
        ))

    return attn_weight, attn_distro, chunk_reduced_value

def forward_core(query, key, value, query_chunk_size, key_chunk_size,
                 mask=None, offset_matrix=None, query_r=None, key_r=None,
```

```
735                      offset_clip_range=128):
736          reduced_values = []
737          all_attn_weights = []
738          mask_value = torch.finfo(query.dtype).min
739          with (torch.no_grad()):
740              for i in range(0, query.size(2), query_chunk_size):
741                  cumu_reduced_chunk = None
742                  cumu_attn_weight = None
743                  query_chunk = query[:, :, i:i + query_chunk_size, :]
744                  attn_weights_ = []
745                  if mask is not None:
746                      mask_qchunk = mask[:, i:i + query_chunk_size, :].to(query_chunk.device)
747                  else:
748                      mask_qchunk = None
749                  if offset_matrix is not None:
750                      offset_qchunk = offset_matrix[:, i:i + query_chunk_size, :].to(query_chunk.device)
751                  else:
752                      offset_qchunk = None
753
754                  for j in range(0, key.size(2), key_chunk_size):
755                      key_chunk = key[:, :, j:j + key_chunk_size, :]
756                      value_chunk = value[:, :, j:j + key_chunk_size, :]
757                      if mask is not None:
758                          mask_chunk = mask_qchunk[:, :, j:j + key_chunk_size]
759                      else:
760                          mask_chunk = None
761                      if offset_matrix is not None:
762                          offset_chunk = offset_qchunk[:, :, j:j + key_chunk_size]
763                      else:
764                          offset_chunk = None
765                      attn_weight, _, reduced_chunk = \
766                      attn_core_logsumexp(query_chunk, key_chunk, value_chunk,
767                              mask_chunk=mask_chunk, offset_chunk=offset_chunk,
768                              query_r=query_r, key_r=key_r,
769                              offset_clip_range=offset_clip_range
770                              )
771
772                      if cumu_reduced_chunk is None:
773                          cumu_reduced_chunk = reduced_chunk.to(torch.float32)
774                          cumu_attn_weight = attn_weight
775                      else:
776                          cumu_attn_weight = torch.stack([cumu_attn_weight, attn_weight])
777                          cumu_reduced_chunk = torch.stack([cumu_reduced_chunk, reduced_chunk])
778                          cumu_reduced_chunk = torch.einsum(
779                              "tbnqd,tbnq->bnqd",
780                              cumu_reduced_chunk.to(torch.float32),
781                              cumu_attn_weight.softmax(dim=0)
782                          ).to(torch.float32)
783                          cumu_attn_weight = cumu_attn_weight.logsumexp(dim=0)
784                      attn_weights_.append(attn_weight)
785                  reduced_values.append(cumu_reduced_chunk.to(query_chunk.dtype))
786                  all_attn_weights.append(cumu_attn_weight)
787              reduced_values = torch.cat(reduced_values, dim=2)
788
789              all_attn_weights = torch.cat(all_attn_weights, dim=2)
790
791              return reduced_values, all_attn_weights
792
793  def backward_core(grad_output, query, key, value, all_attn_weights, chunk_size,
794                    mask=None, offset_matrix=None, query_r=None, key_r=None,
795                    offset_clip_range=128):
796      grad_query = torch.zeros_like(query, dtype=torch.float32)
797      grad_key = torch.zeros_like(key, dtype=torch.float32)
798      grad_value = torch.zeros_like(value, dtype=torch.float32)
799      grad_query_r = torch.zeros_like(query_r, dtype=torch.float32) if query_r is not None else None
800      grad_key_r = torch.zeros_like(key_r, dtype=torch.float32) if key_r is not None else None
801      scale = 1.0 / math.sqrt(value.shape[-1])
802
803      with torch.no_grad():
804          for i in range(0, query.size(2), chunk_size):
805              query_chunk = query[:, :, i:i + chunk_size, :].contiguous()
806              attn_weights_chunk = all_attn_weights[:, :, i:i + chunk_size].contiguous()
807              grad_output_chunk = grad_output[:, :, i:i + chunk_size, :].contiguous()
808
809              if mask is not None:
810                  mask_qchunk = mask[:, i:i + chunk_size, :].to(query.device)
811              else:
812                  mask_qchunk = None
813              if offset_matrix is not None:
814                  offset_qchunk = offset_matrix[:, i:i + chunk_size, :].to(query.device)
815              else:
816                  offset_qchunk = None
817
818              accumu_modifier = 0.
819
820              for j in range(0, key.size(2), chunk_size):
821                  key_chunk = key[:, :, j:j + chunk_size, :]
822                  value_chunk = value[:, :, j:j + chunk_size, :]
823
824                  if mask is not None:
825                      mask_chunk = mask_qchunk[:, :, j:j + chunk_size]
826                  else:
```

```
827                 mask_chunk = None
828             if offset_matrix is not None:
829                 offset_chunk = offset_qchunk[:, :, j:j + chunk_size]
830             else:
831                 offset_chunk = None
832
833             # Forward pass to recompute intermediates
834             attn_weight, attn_distro, reduced_chunk = attn_core_logsumexp(
835                 query_chunk, key_chunk, value_chunk,
836                 mask_chunk=mask_chunk, offset_chunk=offset_chunk,
837                 query_r=query_r, key_r=key_r,
838                 offset_clip_range=offset_clip_range
839             )
840
841             adjustment = (attn_weight - attn_weights_chunk).exp().unsqueeze(dim=-1)
842             attn_global = attn_distro * adjustment
843
844             grad_attn_global = torch.einsum(
845                 "bnqd,bnkd->bnqk", grad_output_chunk, value_chunk
846             )
847             grad_value_chunk = torch.einsum('bnqk,bnqd->bnkd',
848                                             attn_global.to(grad_output_chunk.dtype),
849                                             grad_output_chunk)
850             grad_value[:, :, j:j + chunk_size, :] += grad_value_chunk.to(torch.float32)
851
852             accumu_modifier -= torch.einsum("bnqk,bnqk->bnq",
853                                             grad_attn_global.to(torch.float32),
854                                             attn_global).unsqueeze(dim=-1)
855
856         for j in range(0, key.size(2), chunk_size):
857             key_chunk = key[:, :, j:j + chunk_size, :]
858             value_chunk = value[:, :, j:j + chunk_size, :]
859
860             if mask is not None:
861                 mask_chunk = mask_qchunk[:, :, j:j + chunk_size]
862             else:
863                 mask_chunk = None
864             if offset_matrix is not None:
865                 offset_chunk = offset_qchunk[:, :, j:j + chunk_size]
866             else:
867                 offset_chunk = None
868
869             # Forward pass to recompute intermediates
870             attn_weight, attn_distro, reduced_chunk = attn_core_logsumexp(
871                 query_chunk, key_chunk, value_chunk,
872                 mask_chunk=mask_chunk, offset_chunk=offset_chunk,
873                 query_r=query_r, key_r=key_r,
874                 offset_clip_range=offset_clip_range
875             )
876
877             adjustment = (attn_weight - attn_weights_chunk).exp().unsqueeze(dim=-1)
878             attn_global = attn_distro * adjustment
879
880             grad_attn_global = torch.einsum(
881                 "bnqd,bnkd->bnqk", grad_output_chunk, value_chunk
882             )
883
884             grad_attn_scores = attn_global * (
885                 grad_attn_global + accumu_modifier
886             )
887
888             # Compute gradients w.r.t. query and key
889             grad_attn_scores_qk = (grad_attn_scores * scale).to(query_chunk.dtype)
890
891             grad_query_chunk = torch.einsum('bnqk,bnkd->bnqd', grad_attn_scores_qk, key_chunk)
892             grad_key_chunk = torch.einsum('bnqk,bnqd->bnkd', grad_attn_scores_qk, query_chunk)
893
894
895             # If query_r and key_r are used, compute their gradients
896             if offset_chunk is not None and query_r is not None and key_r is not None:
897                 offset_chunk = offset_chunk.clamp(min=-offset_clip_range, max=offset_clip_range)
898                 indexed_offset = offset_chunk + offset_clip_range
899                 indexed_offset = indexed_offset.unsqueeze(1).expand(-1, query_r.size(1), -1, -1)
900
901                 # Gradients w.r.t. attn_scores_qbias and attn_scores_kbias
902                 grad_attn_scores_qbias = grad_attn_scores_qk.to(torch.float32)
903                 grad_attn_scores_kbias = grad_attn_scores_qk.to(torch.float32)
904
905                 # Ungather gradients
906                 grad_attn_scores_qbias_full = torch.zeros(
907                     grad_attn_scores_qbias.size(0),
908                     grad_attn_scores_qbias.size(1),
909                     query_r.size(0),
910                     grad_attn_scores_qbias.size(3),
911                     device=grad_attn_scores_qbias.device,
912                     dtype=torch.float32
913                 ).scatter_add_(2, indexed_offset, grad_attn_scores_qbias)
914
915                 grad_attn_scores_kbias_full = torch.zeros(
916                     grad_attn_scores_kbias.size(0),
917                     grad_attn_scores_kbias.size(1),
918                     grad_attn_scores_kbias.size(2),
```

```
919                          key_r.size(0),
920                          device=grad_attn_scores_kbias.device,
921                          dtype=torch.float32
922                      ).scatter_add_(3, indexed_offset, grad_attn_scores_kbias)
923
924                      # Compute gradients w.r.t. query_r and key_r
925                      grad_query_r += torch.einsum('bnik,bnkd->ind', grad_attn_scores_qbias_full,
926                          key_chunk.to(torch.float32))
927                      grad_key_chunk += torch.einsum('bnik,ind->bnkd', grad_attn_scores_qbias_full,
928                          query_r.to(torch.float32))
929
930                      grad_key_r += torch.einsum('bnqj,bnqd->jnd', grad_attn_scores_kbias_full,
931                          query_chunk.to(torch.float32))
932                      grad_query_chunk += torch.einsum('bnqj,jnd->bnqd', grad_attn_scores_kbias_full,
933                          key_r.to(torch.float32))
934
935                  grad_query[:, :, i:i + chunk_size, :] += grad_query_chunk.to(torch.float32)
936                  grad_key[:, :, j:j + chunk_size, :] += grad_key_chunk.to(torch.float32)
937
938      return grad_query, grad_key, grad_value, grad_query_r, grad_key_r
939
940
941
942
943  class MemoryEfficientAttention(torch.autograd.Function):
944      @staticmethod
945      def forward(ctx, query, key, value, mask=None, offset_matrix=None, query_r=None, key_r=None,
946           chunk_size=1024):
947          dtype = value.dtype
948          # query, key, value = query, key, value
949          key_chunk_size = min(chunk_size * 2, key.size(2))
950          query_chunk_size = min(chunk_size, query.size(2))
951
952          with torch.no_grad():
953              reduced_values, all_attn_weights = forward_core(query, key, value,
954                                                              query_chunk_size, key_chunk_size,
955                                                              mask, offset_matrix,
956                                                              query_r, key_r
957                                                              )
958
959          # Save full query, key, and value tensors, but not the intermediates
960          ctx.save_for_backward(query, key, value, mask, offset_matrix, all_attn_weights, query_r, key_r)
961          ctx.chunk_size = chunk_size
962
963          return reduced_values.to(value.dtype)
964
965      @staticmethod
966      def backward(ctx, grad_output):
967
968          dtype = grad_output.dtype
969          query, key, value, mask, offset_matrix, all_attn_weights, query_r, key_r = ctx.saved_tensors
970          chunk_size = ctx.chunk_size
971
972          # Initialize gradients
973          with (torch.no_grad()):
974              grad_query, grad_key, grad_value, grad_query_r, grad_key_r = \
975              backward_core(grad_output, query, key, value,
976                            all_attn_weights, chunk_size,
977                            mask=mask, offset_matrix=offset_matrix,
978                            query_r=query_r, key_r=key_r)
979
980          return grad_query.to(dtype), grad_key.to(dtype), grad_value.to(dtype), None, None, grad_query_r
981              , grad_key_r, None
982
983  def memory_efficient_attention(query, key, value, mask=None,
984                                 offset_matrix=None, query_r=None, key_r=None,
985                                 chunk_size=2048):
986      return MemoryEfficientAttention.apply(query, key, value, mask,
987                                            offset_matrix, query_r, key_r,
988                                            chunk_size)
989
```

# C  Limitations

## C.1  Data Processing and Language

As is briefly admitted in the main text of the paper, the permutation of insertion operations described in the paper assumes the major proportion of the data is natural language, and mostly implemented to adapt to natural languages with latin-alphabet writing system like English, French and German. For languages with multi-byte-multi-token characters like Japanese and Chinese, current preprocessing pipeline fall back to autoregressive/identical permutation. This allows the toleration of moderately mixed-in multiligual data, but is definitely far from being the optimal solution. We leave this for future work.

## C.2  Broader Impact

As a generative model with better controllability than current autoregressive models, due to the inevitable data bias, pretrained InsNeXts are at risk of producing harmful contents that may offend people of different self-identification and/or play a negative role in misinformation spreading. We have only scaled the model to be limited sizes as large as the maximum of 0.6B parameters, so this is currently still safely contained. However, we definitely call for regarding a broader societal impact when further scaling this proposed architecture in the future.

## C.3  Suboptimal Hardware-oriented Optimization

We acknowledge our limitation of ability in pushing the hardware-related optimization to the best practice like in FlashAttention-2/3. According to our statistics, given the same model scale, our training efficiency is still inferior compared to LLMs with mature architectures like LLaMA. We look forward to broader collaboration to solve this issue in the future.

