# OpenReview forum: "InsNeXt: Training Scalable Insertion-based Language Models from Scratch"
_NeurIPS.cc/2025/Conference — Submitted to NeurIPS 2025_

### Official Review · Reviewer_gHuc · 2025-07-03

**Clarity:** 3
**Significance:** 2
**Originality:** 2
**Rating:** 4
**Confidence:** 1

**Summary:**

This paper presents InsNeXt, a new scalable insertion-based language model that integrates recent innovations from large language model training (e.g., efficient attention, improved normalization, and flexible position encodings). The work addresses key inefficiencies in previous insertion-based models such as InsNet, proposing novel techniques like efficient bidirectional re-contextualization and soft-capped position logits. The authors train models from 154M to 573M parameters and evaluate them on natural language understanding (GLUE), commonsense reasoning (HellaSwag, WinoGrande, OBQA), and controllable generation tasks (CommonGen, Yelp). The proposed model demonstrates strong controllability and competitive performance compared to autoregressive baselines.

**Questions:**

1. The proposed bidirectional re-contextualization mechanism aims to mitigate exposure bias while maintaining efficiency. Can the authors provide quantitative measurements of how often re-encoding is triggered during inference and its relative impact on decoding latency and throughput?
2. InsNeXt still underperforms RoBERTa and BERT-large on NLU tasks like MNLI. What are the authors’ hypotheses on this gap? Is it due to the insertion paradigm itself, or could further scaling or architectural tuning (e.g., attention heads or deeper layers) close this gap?

**Ethical Concerns:**

["NO or VERY MINOR ethics concerns only"]

**Final Justification:**

I think the rebuttal addresses my concerns so I raise my score. However, my confidence is quite low for this paper since it might lie outside my main research field.

**Limitations:**

yes

**Quality:**

2

**Strengths And Weaknesses:**

Honestly, I am not an expert in this field, but I would like to give some comments based on my understanding and background knowledge:

Strengths:
- Presents a modern and scalable insertion-based LM, which is timely given growing interest in controllable and efficient generation.
- Proposes methodologically novel components, such as the bidirectional re-contextualization strategy and soft-capped position logits, addressing known limitations in insertion-based models.
- Offers extensive empirical evaluation across NLU, reasoning, and generation tasks, with comparisons to autoregressive and encoder-only models. Provides clear architectural ablations and implementation details that contribute to reproducibility.

Weaknesses:
- While the model shows improvements over prior insertion models, its representation quality still lags behind SOTA encoder-based models like RoBERTa, limiting its applicability in high-performance NLU tasks.
- The re-contextualization mechanism, though efficient in expectation, introduces computational spikes and its cost is not fully quantified.

---

> ### Author Rebuttal · Authors · 2025-07-30
>
> We thank the reviewer for feedback. We hereby try to respond to each point of them one by one:
>
>  - Re Weakness 1) and Question 2) **”Representation quality falling short against BERT-style works”** We acknowledge that the proposed InsNeXt model falls short against encoder-only models like RoBERTa-large on NLU tasks. We hypothesize that this is due to multiple reasons. First, as all existing generative models including the more recent ones like Qwen falling short against RoBERTa, we actually outperform all autoregressive LMs. We argue that there might be some inherent limitations of the representation produced by generative models, possibly due to how information shall be differently compressed for generation/understanding. Second, RoBERTa is trained for 40Epochs on 160GB text data, which is approximately 1720B tokens, whereas the total number of trained tokens in InsNeXt can barely reach 180B tokens. In fact, we still observe steady performance gains in InsNeXt with continued training (either with more epochs on the same data we already sampled or sampling more from SlimPajama). We are unable to proceed due to the limitation of our computational resources. We respectfully ask for the reviewer’s understanding.
>
> - Re Weakness 2) and Question 1) **”Inference efficiency concerns with Efficient Bidirectional Recontextualization”**  We hereby try to both theoretically and practically analyze the performance w/ the proposed Efficient Bidirectional Recontextualization. As is cited in our paper, we refer to [1] for the proof that, the amortised cost of each insertion operation w/ EBR is O(1 + T(n) / n), where T(n) is the cost for the transformer model to bidirectionally encode an n-length sequence. According to reports from FlashAttention [2, Figure 3], with modern computational infrastructures and efficient attention kernels, T(n) is empirically sublinear to at most linear w.r.t. n when n < 8K. This reflects the fact that the major bottleneck in this case is the tensor copying overhead, which matches the original assumptions in C++ Vector container’s proof. To effectively generalize this assumption to longer context, in practice we can and should always partially bidirectionally re-contextualize within the most recent 8K token window. The following is a more self-contained proof for **EBR’s amortised per-insertion cost being O(1)**:
>
> **Assumption**: 1) Each newly processed token causes a uniformly distributed copying cost $\eta$;
> 2) Within the concerned range of context length,  the cost for the transformer model to bidirectionally encode an n-length sequence T(n) is O(n);
>
> **Proof.** Without loss of generality, we consider a growing sequence from 1 to $n = 2^m$ tokens.
>
> The total number of copying cost induced by these $n$ insertions is $O(n\cdot\eta)=O(n)$.
>
> Amongst these insertions, at the steps $i = 2, 4, 8, …, 2^m$, the spiking bidirectional recontextualization causes a cost of $O(2 + 4 + 8 +... +2^m) = O(2^{m+1}-2) = O(2n) = O(n)$
>
> The total overhead cost is thus O(n). Divided by the number of steps $n$, we get O(1) as the amortised cost per-step.
>
> It is trivial to prove the general case, since a growing sequence ending with a non-spiking step simply adds linear copying cost which does not violate the O(1) complexity.
>
> Empirically, in our experiments where the generation procedures do not exceed 128 tokens, we do not observe a statistically significant overhead from the proposed EBR. On a single RTX4090 GPU hosted on WSL 2, Ubuntu 22.04 and the huggingface transformers library, with our 0.6B model, a trivial step that does not trigger the recontextualization takes approximately 13.3ms on average. The spiking step that triggers it takes approximately 21.3ms, including all time cost to update the recontextualized QK caches. For longer context, this additional overhead can always be contained in the current magnitude by using a local window, such that we only re-contextualize the nearest, say 128 or 1024 tokens. The user also get to customize if they want to resize the window or trigger the recontextualization less/more often for better efficiency/quality.
>
> [1] Thomas H. Cormen, Charles E. Leiserson, Ronald L. Rivest, and Clifford Stein. 2022. Dynamic tables, 4 edition, chapter 17. MIT Press, Cambridge, MA. Section 17.4.
>
> [2] Tri Dao, Daniel Y. Fu, Stefano Ermon, Atri Rudra, and Christopher Ré. 2022. Flash Attention: Fast and memory-efficient exact attention with io-awareness. arXiv preprint arXiv:2205.14135. Spotlight paper at ICLR 2023.
>
> We appreciate the reviewer’s suggestions, and look forward to further discussion. We respectfully ask the reviewer if you can re-evaluate our work, should the major concerns be addressed here or in our extended conversation.

---

> > ### Comment · Reviewer_gHuc · 2025-08-02
> > **Thanks for the rebuttal**
> >
> > Thanks for the detailed response. I think it addresses my concerns and I will raise my score.

---

### Official Review · Reviewer_zq4E · 2025-07-03

**Clarity:** 3
**Significance:** 3
**Originality:** 3
**Rating:** 4
**Confidence:** 5

**Summary:**

This work proposed InsNeXt that integrates the recent advancement of LM to achieve improved scalability. The InsNeXt is designed to overcome the training inefficiencies and architectural limitations of earlier insertion-based approaches (e.g., InsNet). The authors build upon and improve prior work by integrating modern LLM advances such as efficient attention mechanisms (e.g., FlexAttention), improved normalization (e.g., pre-LN, DeepNorm), and optimized training pipelines. The InsNeXt is scaled from 154M to 0.6B parameters. Furthermore, a novel context encoding mechanism is proposed for insertion-based decoding. During the inference time, a sparse re-encoding mechanism is proposed. The evaluation on tasks such as representation learning, commonsense reasoning, and controllable generation show that the InsNeXt models are competitive as the state-of-the-art autoregressive models with similar size.

**Questions:**

- Any statistics on the training/inference efficiency?
- Do you think if there are any challenges (algorithm aspect or training aspect) when the model size is further scaled up?

**Ethical Concerns:**

["NO or VERY MINOR ethics concerns only"]

**Final Justification:**

The author explained the performance gap against BERT-style work and provide more analysis about the efficiency.
These efforts are greatly appreciated. However, Authors did not address the concerns about scalability (4aqz also raised the issue), I will keep the score as borderline accept.

**Limitations:**

Yes

**Quality:**

3

**Strengths And Weaknesses:**

### Strengths
- I would acknowledge the effort of thorough ablation studies conducted in this work. The paper details an exhaustive ablation of model components to reveal good practices to the research community.
- One novelty brought by the paper is the efficient bidirectional re-contextualization, that addresses exposure bias and better handling the user-input prompt.
- The model is tested across representation, reasoning, and generation, offering a well-rounded assessment of its strengths. The InsNeXt-base model outperforms its baseline InsNext.
### Weaknesses
- Performance itself is still a concern to me. While it is competitive, InsNeXt does not match SOTA discriminative models like RoBERTa-base (when compared with InsNeXt-base) and RoBERTa-large (when compared with InsNeXt-advanced) in representation learning, suggesting some architectural constraints in my mind. But I did acknowledge the model performance on other task categories.
- While position encoding is a core design point (InsNet-style sinusoidal vs. insertion-aligned ALiBi), the paper lacks direct side-by-side comparisons across tasks and scales. It's unclear how each encoding affects models e.g. generalization etc.

---

> ### Author Rebuttal · Authors · 2025-07-30
>
> We thank the reviewer for the thorough reading and comprehensive feedback. We hereby try to respond to each point of them one by one:
>
>  - Re Weakness 1) **"Representation quality falling short against BERT-style works"** We acknowledge that the proposed InsNeXt model falls short against encoder-only models like RoBERTa-large on NLU tasks. We hypothesize that this is due to multiple reasons. First, as all existing generative models including the more recent ones like Qwen falling short against RoBERTa, we actually outperform all autoregressive LMs. We argue that there might be some inherent limitations of the representation produced by generative models, possibly due to how information shall be differently compressed for generation/understanding. Second, RoBERTa is trained for 40Epochs on 160GB text data, which is approximately 1720B tokens, whereas the total number of trained tokens in InsNeXt can barely reach 180B tokens. In fact, we still observe steady performance gains in InsNeXt with continued training (either with more epochs on the same data we already sampled or sampling more from SlimPajama). We are unable to proceed due to the limitation of our computational resources. We respectfully ask for the reviewer’s understanding.
>
> - Re Weakness 2) and Question 1) **"Direct comparison between position encodings, Training and inference efficiency"**  We respectfully ask the reviewer to refer to Table 4 in our appendix for a direct comparison between position encodings in representation learning performance and computational efficiency. In fact, due to the huge scalability differences between the possible choices, all other position encodings than the proposed insertion-oriented ALiBi are practically too slow to be trained on the longer sequences from SlimPajama.
>
>   For inference efficiency, we hereby try to both theoretically and practically analyze the performance w/ the proposed Efficient Bidirectional Recontextualization. As is cited in our paper, we refer to [1] for the proof that, the amortised cost of each insertion operation w/ EBR is O(1 + T(n) / n), where T(n) is the cost for the transformer model to bidirectionally encode an n-length sequence. According to reports from FlashAttention [2, Figure 3], with modern computational infrastructures and efficient attention kernels, T(n) is empirically sublinear to at most linear w.r.t. n when n < 8K. This reflects the fact that the major bottleneck in this case is the tensor copying overhead, which matches the original assumptions in C++ Vector container’s proof. To effectively generalize this assumption to longer context, in practice we can and should always partially bidirectionally re-contextualize within the most recent 8K token window. The following is a more self-contained proof for **EBR’s amortised per-insertion cost being O(1)**:
>
> **Assumption**: 1) Each newly processed token causes a uniformly distributed copying cost $\eta$;
> 2) Within the concerned range of context length,  the cost for the transformer model to bidirectionally encode an n-length sequence T(n) is O(n);
>
> **Proof.** Without loss of generality, we consider a growing sequence from 1 to $n = 2^m$ tokens.
>
> The total number of copying cost induced by these $n$ insertions is $O(n\cdot\eta)=O(n)$.
>
> Amongst these insertions, at the steps $i = 2, 4, 8, …, 2^m$, the spiking bidirectional recontextualization causes a cost of $O(2 + 4 + 8 +... +2^m) = O(2^{m+1}-2) = O(2n) = O(n)$
>
> The total overhead cost is thus O(n). Divided by the number of steps $n$, we get O(1) as the amortised cost per-step.
>
> It is trivial to prove the general case, since a growing sequence ending with a non-spiking step simply adds linear copying cost which does not violate the O(1) complexity.
>
> Empirically, in our experiments where the generation procedures do not exceed 128 tokens, we do not observe a statistically significant overhead from the proposed EBR (<30ms/per step). A single decoding step with QK cache for InsNeXt also resembles that of a plain autoregressive language model.
>
>
> [1] Thomas H. Cormen, Charles E. Leiserson, Ronald L. Rivest, and Clifford Stein. 2022. Dynamic tables, 4 edition, chapter 17. MIT Press, Cambridge, MA. Section 17.4.
>
> [2] Tri Dao, Daniel Y. Fu, Stefano Ermon, Atri Rudra, and Christopher Ré. 2022. Flashattention: Fast and memory-efficient exact attention with io-awareness. arXiv preprint arXiv:2205.14135. Spotlight paper at ICLR 2023.
>
> We appreciate the reviewer’s many constructive comments and feedback, and look forward to further discussion. We respectfully ask the reviewer if you can further acknowledge our work, should the concerns be addressed here or in our extended conversation.

---

### Official Review · Reviewer_8KMZ · 2025-07-07

**Clarity:** 2
**Significance:** 2
**Originality:** 2
**Rating:** 4
**Confidence:** 4

**Summary:**

This paper proposes an insertion based model called InsNeXt, which is improved based on InsNet. For training, it combines sentence-level pretraining (on Wikipedia and books dataset) and document-level pretraining (on SlimPajama dataset). For inference, it uses a proposed bidirectional context re-encoding mechanism, which keeps a bidirectional encoded part and only use unidirectional updates until the generated tokens surpass the bidirectional part.

**Questions:**

Please prove that the expected computational overhead of a decoding process with the proposed re-contextualization is at the same scale of the vanilla uni-directional one.

**Ethical Concerns:**

["NO or VERY MINOR ethics concerns only"]

**Final Justification:**

The authors have done a considerable amount of work to replace components of InsNet with more efficient and scalable methods, including flash attention, insertion-oriented ALiBi, and etc. Although the contribution in engineering might overweigh the contribution in novelty, a more modern and scalable insertion-based model is necessary to the community. Hence, I will give a borderline accept.

**Limitations:**

Yes

**Paper Formatting Concerns:**

No.

**Quality:**

3

**Strengths And Weaknesses:**

Strength
1. This paper integrates components and methods from autoregressive LLMs to insertion-based models, so it is more scalable than InsNet.
2. They have done study on various alternative designs for different components of insertion-based models to select the best solutions.
3. The experimental and model details are clearly stated in the paper.


Weakness
1. Most important works, such as overcoming the volatile position problem, has been done by InsNet. The main innovative contribution is Efficient Bidirectional Re-contextualization in 2.4, but it is simple and lacks proof. Please add more explanation in 2.4.
2. In Section 2, revisiting of the existing InsNet, ablation study on alternative design details, and the novel proposed idea are all included. Please make them separate sections.
3. It is claimed that this model has better computational efficiency, but no theoretical proof or experimental numbers is shown.
4. Insertion-based models can potentially be more efficient than autoregressive models in specific tasks, such as code editing, but its special feature is not well utilized to design a novel efficient method.

---

> ### Author Rebuttal · Authors · 2025-07-30
>
> We thank the reviewer for the suggestions. We hereby try to respond to each point of them one by one:
>
>  - Re Weakness 1a) **”Limited novelty”** Just to clarify, the main contribution of this paper focuses on tackling the training-time efficiency problem of insertion-based language models. While we acknowledge that InsNet has addressed many major efficiency issues of insertion-based language models, it only considers the case where **training sequences are short, thus no efficient attention mechanism is needed**. The original sinusoidal position encoding in InsNet falls obsolete and is **incompatible** with more recently studied efficient attention kernels like FlashAttn and FlexAttn. We’ve tackled these problems with InsNeXt. To the best of our knowledge, we are the first to scale up fully insertion-based language models. By showing the community some promising results on scaling up insertion-based models, we want to attract more attention and resources to further boost the scaling-up of such a new paradigm of language models.
>
>  - Re Weakness 1b) and Question 1 **”Proof of correctness of EBR”**: As is cited in our paper, we refer to [1] for the proof that, the amortised cost of each insertion operation w/ EBR is O(1 + T(n) / n), where T(n) is the cost for the transformer model to bidirectionally encode an n-length sequence. According to reports from FlashAttention [2, Figure 3], with modern computational infrastructures and efficient attention kernels, T(n) is empirically sublinear to at most linear  w.r.t. n when n < 8K. This reflects the fact that the major bottleneck in this case is the tensor copying overhead, which matches the original assumptions in C++ Vector container’s proof. To effectively generalize this assumption to longer context, in practice we can and should always partially bidirectionally re-contextualize within the most recent 8K token window. The following is a more self-contained proof for **EBR’s amortised per-insertion cost being O(1)** that we plan to add to our revision:
>
> **Assumption**: 1) Each newly processed token causes a uniformly distributed copying cost $\eta$;
> 2) Within the concerned range of context length, the cost for the transformer model to bidirectionally encode an n-length sequence T(n) is O(n);
>
> **Proof.** Without loss of generality, we consider a growing sequence from 1 to $n = 2^m$ tokens.
>
> The total number of copying cost induced by these $n$ insertions is $O(n\cdot\eta)=O(n)$.
>
> Amongst these insertions, at the steps $i = 2, 4, 8, …, 2^m$, the spiking bidirectional recontextualization causes a cost of $O(2 + 4 + 8 +... +2^m) = O(2^{m+1}-2) = O(2n) = O(n)$
>
> The total overhead cost is thus O(n). Divided by the number of steps $n$, we get O(1) as the amortised cost per-step.
>
> It is trivial to further prove the general case, since a growing sequence ending with a non-spiking step simply adds linear copying cost which does not violate the O(1) complexity.
>
> [1] Thomas H. Cormen, Charles E. Leiserson, Ronald L. Rivest, and Clifford Stein. 2022. Dynamic tables, 4 edition, chapter 17. MIT Press, Cambridge, MA. Section 17.4.
>
> [2] Tri Dao, Daniel Y. Fu, Stefano Ermon, Atri Rudra, and Christopher Ré. 2022. Flashattention: Fast and memory-efficient exact attention with io-awareness. arXiv preprint arXiv:2205.14135. Spotlight paper at ICLR 2023.
>
>  - Re Weakness 2) **”Writing Restructuring Suggestions”** We thank the reviewer for the writing suggestions. We will update the draft accordingly.
>
>  - Re Weakness 3) **”Need Training Efficiency Proof”** We respectfully ask the reviewer to refer to Table 4 in our appendix. In this table, we show that by adopting the proposed insertion-oriented ALiBi, the insertion-based language model can be trained $6.68x$ faster compared to InsNet, resulting in 116K tokens/second/Chip for the $base$-sized model at the context length of 1024 tokens. This advantage can only grow with a longer context/larger model as insertion-oriented ALiBi is the only one practically compatible with FlexAttn (a customizable FlashAttn alternative). Please let us know if you have further concerns.
>
>  - Re Weakness 4) **”Code editing experiments”** We agree that code editing is one interesting task that insertion-based language models should have a good advantage over traditional autoregressive models. However, in this paper we want to establish the foundation of scalability for fully insertion-based language models, following how autoregressive models like GPT were originally developed. We thank the reviewer again for this suggestion as we will definitely put high priority on this direction in our future work.
>
> We appreciate the reviewer’s many constructive comments and feedback, and look forward to further discussion. We respectfully ask the reviewer if you can re-evaluate our work, should the major concerns be addressed here or in our extended conversation.

---

> > ### Comment · Reviewer_8KMZ · 2025-08-06
> >
> > It looks like the efficiency improvement in Table 4 is showing the advantage of the positional encoding method (insertion-oriented ALiBi) with InsNet, not the Efficient Bidirectional Re-contextualization with InsNeXT? Do you have more evidence for efficiency improvement?
> > As other reviewers mentioned, showing the improvement on efficiency is important.

---

> > > ### Author Response · Authors · 2025-08-06
> > >
> > > Also, just to clarify any potential confusions in the general understanding of our core contributions, we didn't particularly claim the general inference-time efficiency in this work. We focus on the training efficiency and scalability of such insertion-based language models.

---

> > > > ### Comment · Reviewer_8KMZ · 2025-08-07
> > > >
> > > > I have seen the theoretical proof above. I mean can you show the improvement of training efficiency in practice? Similar to training throughput in Table 4 but for Efficient Bidirectional Re-contextualization with InsNeXT. Using other metrics (training time, GPU usage, number of computation, etc) is also fine.

---

> > > > > ### Author Response · Authors · 2025-08-07
> > > > >
> > > > > There might be a major misunderstanding here. What we meant above is that Efficient Bidirectional Re-contextualization is an inference-only algorithm that is not directly involved during training.

---

> > > > > > ### Comment · Reviewer_8KMZ · 2025-08-07
> > > > > >
> > > > > > Sorry for the confusion. Because you mention you only focus on training efficiency, not inference efficiency, I forget Efficient Bidirectional Re-contextualization is only for inference.
> > > > > > The reason why we ask for efficiency is that the accuracy on experiment 1&2 is lower than SoTA. Hence you need to show the advantages of the proposed model. My understanding is that most of the advantages shown in the paper are from combining InsNet and Flash attention. Therefore, I hope you can show more experimental metrics for improvement that is brought by your proposed method, such as Efficient Bidirectional Re-contextualization or modified insertion-oriented ALiBi.

---

> > > > > > > ### Author Response · Authors · 2025-08-07
> > > > > > >
> > > > > > > Thank you for your understanding. As is described in the paper, we show that it is **impossible to trivially combine the vanilla InsNet with FlashAttention**, as it violates the basic computational assumption of the algorithm that all information (semantic and position) should be fully stored in Q,K and V. Block-wise computed memory efficient attention kernels like MemEffAttn and FlashAttn mostly rely on this assumption such that they can effectively partition the computation of attention both in forward pass and backward pass.
> > > > > > >
> > > > > > > In vanilla InsNet, they compute the insertion-oriented relative position encoding using the XLNet-style sinusoidal R matrix that maintains its own linear transformation weight $W_R$. This caused the gradient computation to be mostly non-partitionable in the backward pass. We refered to Table 4 to show that, even at our best efforts with our Triton-implemented MemEffAttn for it (which still successfully fulfilled the memory-efficiency goal as is proposed in the original paper [1]), vanilla InsNet still suffers from severe inefficiency in computational overhead and is mostly less scalable in practice.
> > > > > > >
> > > > > > > One of the core contribution of our proposed InsNeXt is that we proposed an alternative way to implement the insertion-oriented relative position that is more computation-friendly, i.e. the insertion-oriented ALiBi. We was trying to show you in Table 4 the significance of our proposed insertion-oriented ALiBi against the vanilla InsNet's original design of sinusoidal position encoding in scalability.
> > > > > > >
> > > > > > > For the experimental impact of EBR, it plays mostly no role in NLU tasks and the discriminative commonsense reasoning tasks as the context is encoded in its entirety anyways. If you are asking for an ablation study for its role in generative tasks, below is our original record for the two models w/o EBR:
> > > > > > >
> > > > > > > ________________________________________________________
> > > > > > >  Model	| #Params	| Yelp 160K	| News	| CommonGen-dev
> > > > > > >
> > > > > > > _base_	| 154M	| 6.73/100%	| 6.60/100%	| 23.46/97.9%
> > > > > > >
> > > > > > > -w/o EBR | 154M	| 5.91/100%	| 5.32/100%	| 22.88/98.1%
> > > > > > >
> > > > > > > _advanced_	| 573M	| 8.13/100%	| 8.01/100%	| 25.73/98.7%
> > > > > > >
> > > > > > > -w/o EBR | 573M	| 7.79/100%	| 7.61/100%	| 24.51/98.6%
> > > > > > > _________________________________________________________
> > > > > > >
> > > > > > > We also want to briefly explain the computational overhead. Empirically, there isn't much of an observable computational overhead difference when EBR is enabled.  As is included in some of our other responses, for the step where the spiking recontextualization is triggered, it takes ~21.3ms (including the bidirectional re-encoding and KV-cache replacement). For all decoding steps, the average latency is ~13.3ms. If we completely turn off EBR, due to less conditional execution of logic that might help cache utilization, the average latency is ~12ms on our infrastructure. We will update this ablative result to Table 3.
> > > > > > >
> > > > > > > We hope this further explanation helps addressing the concerns, thank you.
> > > > > > >
> > > > > > > [1] Rabe, Markus N., and Charles Staats. "Self-attention does not need $ O (n^ 2) $ memory." arXiv preprint arXiv:2112.05682 (2021).

---

> > > > > > > > ### Comment · Reviewer_8KMZ · 2025-08-07
> > > > > > > >
> > > > > > > > I have changed the rating.
> > > > > > > > If the insertion-oriented ALiBi is one of core contributions, you should move insertion-oriented ALiBi from appendix to the main section. Please revise your paper and emphasize the core contributions.

---

> > > > > > > > > ### Author Response · Authors · 2025-08-07
> > > > > > > > >
> > > > > > > > > We will definitely do it. Thank you for your time and patience in helping us improve the structuring of the paper.

---

> ### Author Response · Authors · 2025-08-06
>
> **”Proof of correctness and efficiency of EBR”**: As is cited in our paper, we refer to [1] for the proof that, the amortised cost of each insertion operation w/ EBR is O(1 + T(n) / n), where T(n) is the cost for the transformer model to bidirectionally encode an n-length sequence. According to reports from FlashAttention [2, Figure 3], with modern computational infrastructures and efficient attention kernels, T(n) is empirically sublinear to at most linear  w.r.t. n when n < 8K. This reflects the fact that the major bottleneck in this case is the tensor copying overhead, which matches the original assumptions in C++ Vector container’s proof. To effectively generalize this assumption to longer context, in practice we can and should always partially bidirectionally re-contextualize within the most recent 8K token window. The following is a more self-contained proof for **EBR’s amortised per-insertion cost being O(1)** that we plan to add to our revision:
>
> **Assumption**: 1) Each newly processed token causes a uniformly distributed copying cost $\eta$;
> 2) Within the concerned range of context length, the cost for the transformer model to bidirectionally encode an n-length sequence T(n) is O(n);
>
> **Proof.** Without loss of generality, we consider a growing sequence from 1 to $n = 2^m$ tokens.
>
> The total number of copying cost induced by these $n$ insertions is $O(n\cdot\eta)=O(n)$.
>
> Amongst these insertions, at the steps $i = 2, 4, 8, …, 2^m$, the spiking bidirectional recontextualization causes a cost of $O(2 + 4 + 8 +... +2^m) = O(2^{m+1}-2) = O(2n) = O(n)$
>
> The total overhead cost is thus O(n). Divided by the number of steps $n$, we get O(1) as the amortised cost per-step.
>
> It is trivial to further prove the general case, since a growing sequence ending with a non-spiking step simply adds linear copying cost which does not violate the O(1) complexity.
>
> In comparison, trivially bidirectional contextualization at each step will take the average cost of T(n), which starts with constant value in shorter context, growing into O(n) within in medium length of context and asymptotically into $O(n^2)$ when no effective total parallelization shall be possible. In comparison, EBR is indeed theoretically much more efficient, especially as model size/context length grows large.
>
> [1] Thomas H. Cormen, Charles E. Leiserson, Ronald L. Rivest, and Clifford Stein. 2022. Dynamic tables, 4 edition, chapter 17. MIT Press, Cambridge, MA. Section 17.4.
>
> [2] Tri Dao, Daniel Y. Fu, Stefano Ermon, Atri Rudra, and Christopher Ré. 2022. Flashattention: Fast and memory-efficient exact attention with io-awareness. arXiv preprint arXiv:2205.14135. Spotlight paper at ICLR 2023.

---

### Official Review · Reviewer_4aqz · 2025-07-11

**Clarity:** 2
**Significance:** 2
**Originality:** 2
**Rating:** 3
**Confidence:** 3

**Summary:**

This paper introduces InsNeXt, a new insertion-based language model designed to address the scalability limitations of previous models like Insertion Transformer and InsNet during training. Insertion-based models offer advantages over autoregressive models due to their inference-time efficiency and controllability, allowing tokens to be generated at arbitrary positions.

**Questions:**

1. How does InsNeXt perform when faced with extremely long sequences requiring many insertions, or when a high density of insertions needs to be made in a very short span?
2. Are there specific types of text generation where the insertion mechanism might introduce artifacts or degrade coherence?

**Ethical Concerns:**

["NO or VERY MINOR ethics concerns only"]

**Final Justification:**

I appreciate the insightful answers to my questions (Q1 and Q2). The clarification on the model's specific failure modes with long sequences and the comparison of its artifacts to those of autoregressive models are very valuable and add significant clarity to the work.

However, my primary concerns regarding the experimental evaluation (Weakness 1 and 2) persist. While authors have acknowledged the limitations—specifically the lack of quantitative data on inference latency and the insufficient scale to prove scalability—the proposed solution is to add this information to the camera-ready version. Unfortunately, this does not address the lack of empirical evidence in the current manuscript for review.

Therefore, while the rebuttal has been very helpful for my understanding, my assessment of the paper remains unchanged due to the unresolved concerns about the limited evaluation.

**Limitations:**

yes

**Quality:**

2

**Strengths And Weaknesses:**

Strengths:
1. The paper directly tackles the training-time inscalability issue of previous insertion-based models. InsNeXt's ability to scale up to 0.6B parameters and handle larger context windows (4096 tokens) is a substantial improvement, making these models more practical for real-world applications.
2. The adoption and refinement of the offset matrix for positional encoding (building on InsNet) is a strong point. This approach correctly addresses the "volatile position problem" inherent in insertion-based models, ensuring stable and efficient training by preserving previously computed position encodings.

Weakness
1. The paper describes a mechanism designed for efficiency, it seems that the authors do not provide enough quantitative data in the experimental sections to prove that this mechanism doesn't introduce significant, practical inference latency trade-offs, especially compared to models that don't employ such re-encoding or to "conventional autoregressive decoding" as claimed.
2. The experiment is not sufficient and scalable.
    - The involved model size for pretraining is only 0.6B, which cannot verify the scalability of the proposed method.
    - The paper highlights a "two-stage training" process (sentence-level and document-level) as crucial for better context encoding and scalability. However, the paper lacks detailed ablation studies to demonstrate the individual contribution and necessity of each training stage.

---

> ### Author Rebuttal · Authors · 2025-07-30
>
> We thank the reviewer for the suggestions on clarity, training setup and robustness analysis. We hereby try to respond to each point of them one by one:
>
>  - Re Weakness 1) **”Efficiency concerns”** Just to clarify, the main contribution of this paper focuses on tackling the training-time efficiency problem of insertion-based language models. If the reviewer is referring to the inference-time efficiency concerns about the proposed Efficient Bidirectional Re-contextualization algorithm, we acknowledge that it introduces additional overhead compared to plain QK-cached insertion-based/autoregressive decoding. However, for most decoding steps, the proposed decoding does the same thing as QK-cached decoding. Only at the steps where the re-contextualization is triggered (i.e. uni-directionally cached context exceeds the length of bi-directionally encoded ones), an additional spiking computation of bidirectional re-encoding and QK-cache update shall be needed. Note  that as is referred in our paper (Line 255, or see [1]), the **amortized cost** of this proposed algorithm is the same as plain QK-cached decoding. According to the statistics we currently have, there isn’t an observable computational performance overhead from the spiking bidirectional recontextualization in EBR for sequences at the length of around or less than 512 tokens (we don’t have data for longer sequences). Note that we can always make EBR windowed in longer sequences (i.e. only impact the most recent 1024 or less QK caches instead of recontextualizing the entire history) to keep the spiking overhead low. We can add a plot of the computation overhead in the camera-ready version. Thank you again for your insightful comments on this.
>
> [1] Thomas H. Cormen, Charles E. Leiserson, Ronald L. Rivest, and Clifford Stein. 2022. Dynamic tables, 4 edition, chapter 17. MIT Press, Cambridge, MA. Section 17.4.
>
>  - Re Weakness 2a) **”Insufficient scalability”** While we agree that 0.6B is a limited scale compared to most recently published LLMs, as an academic group with limited resources, in this work, we aim to explore and deliver the best practices to train insertion-based LMs that are at similar computational level of scalability of modern decoder-only autoregressive LLMs. We choose the scale of 0.6B to align with the smallest scale of more recently published open-source LLMs like Qwen2.5-0.5B. By showing the community some promising results on an affordably scaled up model, we want to attract more attention and resources to further boost the scaling-up of such a new paradigm of language models. In our experiments, we’ve shown that at a small-to-medium scale, insertion-based language models are indeed showing promises to be more controllable, capable and performant language models that can facilitate the next generation of AI. To the best of our knowledge, we are the first to scale up fully insertion-based language models to this level.
>
>  - Re Weakness 2b) **”Ablation of the two-staged training”** We’ve already presented the results of a model that is trained only on sentence-level data in our appendix (please refer to Table 4, the InsNet w/ insertion oriented ALiBi roughly shows the performance of models trained on the sentence-level only data), and we will re-compile a new section that specifically discusses the ablation and necessity of the two-staged training in the camera-ready version, emphasizing its significance in accelerating convergence and stabilizing training given our limited computation. We appreciate the reviewer’s constructive suggestion on this.
>
>  - Re Question 1) **”Performance with dense insertion operations and longer context”** To clarify, by default InsNeXt always decodes with dense, sequential (which means one token inserted per step) insertion operations that are QK-cached, with or without the previously proposed EBR. If we understand the reviewer correctly, we want to respectfully point out that all current generative results are **already** produced in a fashion that **in a short to medium length of span, fully-dense insertion operations are performed to form a sentence or document**. For generation of longer contents, due to the limited scale of the model that we eventually obtain, we admit that it is unlikely that the models can still produce reasonable output without a performance loss. In fact, according to our follow-up experiments, after the model approaches the length limit during pretraining, it tends more and more to **insert into and expand previously concluded sentences**, rather than **moving on to the next sentences**. This can be partially alleviated if we only allow the model to insert into the **most recent 32 positions window**, but still eventually leads to degenerated performance beyond approximately 4K tokens.
>
>  - Re Question 2) **”Insertion-induced artifacts”** We thank the reviewer for bringing this up! As is stated in our response to Q1, forcibly generating contents with the windowed insertion trick beyond the model’s designed capability can cause incoherence and artifacts, including but not limited to non-commonsensical (illogical subject-action-object events) and non-grammatical (unpaired parenthesis etc.) but less repetitive output. This observation of out-of-domain misbehaviors is interestingly different from that of autoregressive models which in a similar situation usually starts to produce highly repetitive outputs. We will add this to the limitation section, and leave detailed study of this for future work.
>
> We appreciate the reviewer’s many constructive comments and feedback, and look forward to further discussion. We respectfully ask the reviewer if they can re-evaluate our work, should the major concerns be addressed in our extended conversation.

---

### Author Response · Authors · 2025-08-02
**Look forward to further discussions**

Dear Reviewers,

Thank you very much for your valuable feedback. We have done our best to address your main concerns so far and have prepared responses to each of your comments. As we are now approaching the half-point of the discussion period, we hope there will be more opportunities to interact with you, especially if you have any requests for additional results or experiments. We truly appreciate your time and suggestions, and we are eager to keep improving the paper to better meet the conference’s standards.

---

### Comment · Area_Chair_RGZt · 2025-08-06
**Participation in the rebuttal**

Dear reviewers,

Please engage in the discussion with the authors. The discussion period will end in a few days.

Best,
AC

---

### Note · Authors · 2025-08-13

**Dear Reviewers, AC(s) and PC(s),**

Thank you all for your constructive feedback. In particular, we want to express our sincere appreciation to those of you who actively participated in further discussions with us. Your efforts have contributed to the continuous and significant improvement of this work’s quality, giving us greater hope that it meets the conference’s standards for acceptance.

Below is a brief list of the updates that were proposed and implemented during the rebuttal and discussion period:

1. Additional theoretical and empirical analysis, along with basic statistics, to support the efficiency of the proposed Efficient Bi-directional Re-contextualization algorithm;

2. A brief ablation study demonstrating the necessity of the two-stage pre-training;

3. Clarification that the core contribution of this work lies more in training scalability and generation quality, rather than solely inference-time efficiency;

4. Acknowledgement of the limitation in scale and performance compared to the BERT family. We apologize for this shortcoming and respectfully ask for the reviewers’ understanding that our computational resources are limited due to the general constraints in academia;

5. Other minor explanation and responses to hypothetical questions.

We hope we have sufficiently understand and addressed your requests and concerns. Thank you all again for your dedication to this submission and this year's NeurIPS conference. We enjoyed working with you to continuously improve the fairness and academic rigor of the conference and our shared community.

---

### Decision · Program_Chairs · 2025-09-17

**Decision:**

Reject

**Comment:**

This paper introduces InsNeXt, an insertion-based language model that integrates modern architectural and training advancements to improve scalability. This is  a bordeline paper, reviewers acknowledged the value in modernizing this class of models. The discussion was active, with authors providing additional theoretical analysis for EBR's efficiency, ablation results for key components, and context for performance comparisons against SOTA models. The primary reason for rejection is the insufficient empirical evidence to fully substantiate the paper's central claim of scalability. While scaling to 0.6B parameters is a step forward, it is not sufficient to definitively prove the proposed methods' effectiveness at larger scales, a point strongly made by Reviewer 4aqz. The work is promising, but it requires more substantial experimental validation to be accepted at NeurIPS.